# Wide-Ranging Multitool Study of Structure and Porosity of PLGA Scaffolds for Tissue Engineering

**DOI:** 10.3390/polym13071021

**Published:** 2021-03-25

**Authors:** Alexey V. Buzmakov, Andrey G. Dunaev, Yuriy S. Krivonosov, Denis A. Zolotov, Irina G. Dyachkova, Larisa I. Krotova, Vladimir V. Volkov, Andrew J. Bodey, Victor E. Asadchikov, Vladimir K. Popov

**Affiliations:** 1Institute of Photon Technologies of Federal Scientific Research Centre “Crystallography and Photonics” of Russian Academy of Sciences, Pionerskaya 2, Troitsk, 108840 Moscow, Russia; buzmakov@gmail.com (A.V.B.); dunaewan@gmail.com (A.G.D.); yuri.s.krivonosov@yandex.ru (Y.S.K.); zolotovden1985@gmail.com (D.A.Z.); krollar@yandex.ru (L.I.K.); volkicras@mail.ru (V.V.V.); asad@crys.ras.ru (V.E.A.); 2Diamond Light Source, Harwell Oxford Campus, Didcot OX11 0DE, UK; andrew.bodey@diamond.ac.uk

**Keywords:** PLGA scaffolds, supercritical fluid processing, internal structure, porosity, small-angle X-ray scattering, helium pycnometry, mercury intrusion porosimetry, X-ray microtomography

## Abstract

In this study, the nanoscale transformation of the polylactic-co-glycolic acid (PLGA) internal structure, before and after its supercritical carbon dioxide (sc-CO_2_) swelling and plasticization, followed by foaming after a CO_2_ pressure drop, was studied by small-angle X-ray scattering (SAXS) for the first time. A comparative analysis of the internal structure data and porosity measurements for PLGA scaffolds, produced by sc-CO_2_ processing, on a scale ranging from 0.02 to 1000 μm, was performed by SAXS, helium pycnometry (HP), mercury intrusion porosimetry (MIP) and both “lab-source” and synchrotron X-ray microtomography (micro-CT). This approach opens up possibilities for the wide-scale evaluation, computer modeling, and prediction of the physical and mechanical properties of PLGA scaffolds, as well as their biodegradation behavior in the body. Hence, this study targets optimizing the process parameters of PLGA scaffold fabrication for specific biomedical applications.

## 1. Introduction

Nowadays, biocompatible polymer materials, as well as polymer-based structures, play a pivotal role in the intensive research and extensive development of advanced biomedical devices and pharmaceutical formulations [1,2]. Biodegradable polymers, e.g., aliphatic polyesters, such as polylactic acid (PLA), polyglycolic acid (PGA) and their copolymer polylactic-co-glycolic acid (PLGA), which can “dissolve” at specific rates in the human body, are of particular interest for modern biomedical products (suture threads, stents, custom-designed implants, scaffolds for tissue engineering, etc.) [3,4,5] and controlled drug release carrier [6,7] fabrication.

The physical–mechanical properties of aliphatic polyesters are considered to be among the key parameters when choosing them as row materials for specific biomedical applications where they must mimic the mechanical strength of the living tissue targeted to be replaced or regenerated [8]. For example, PLA and PLGA display significant tensile stress (ca. 50 MPa) and favorable ultimate elongation at breakage (from 3% up to 200%, depending on chemical structure and molecular weight distribution) properties, enabling their broad application in the fabrication of products that are designed for use under constant tensile stress and high elongation conditions [9,10].

There are a large number of chemical and structural characteristics that affect the properties of polymeric materials and their behavior under various functional conditions. The chemical composition, molecular weight distribution, degree of polymer crystallinity, and crosslinking determine the melting points and glass transition temperatures, density, stiffness, strength, ductility, and biodegradation of polymeric materials [11].

The biodegradation rate of aliphatic polyesters is an important parameter that determines the conditions of their behavior in the living body and, as a result, the niche of their research and clinical applications, as well as the design of various devices and pharmaceutical formulations [12,13]. The main chemical route for the degradation of these polymers is hydrolysis, which results in the formation of water-soluble monomers and low-molecular-weight oligomers from their respective large macromolecules [14]. There are two major types of such hydrolysis: (1) *heterogeneous*, when degradation of the polymer begins near the surface and slowly spreads downwards; (2) *autocatalytic*, when the produced acid ester monomers and low-molecular-weight oligomers decrease the pH of the internal volume, accelerating polymer decomposition [15]. Autocatalysis is more efficient within a sample, whereas the surface layer of the polymer sample may persist for a long period of time without visible changes. However, autocatalysis may slow down if acidifying agents can easily leave the polymer due to diffusion (e.g., through the pores or void volume) into the outer (water-containing) medium [16]. All these processes lead to a gradual change in the functional properties of polymeric materials and biodegradable polymeric products. To predict the characteristics and parameters of these changes, it is crucially important to identify the major factors and gather all of the available data related to the intrinsic physical–chemical properties of initial polymers that affect such phenomena.

Porosity is one of the key structural parameters of the scaffolds to be considered for tissue engineering constructions (TECs) [17]. These scaffolds, after implantation in the body (or being placed into a bioreactor) play the role of providing a three-dimensional framework for primary cell attachment, growth, and proliferation, permitting native extracellular matrix formation [18]. Their interconnected structure facilitates the transport of the necessary nutrients and removal of the toxic products of cell metabolism. For these specific applications, total porosity alone does not have a direct impact on the features of the scaffold. Pore size and pore interconnectivity are more important. However, in many cases, polymer scaffolds might contain both closed (isolated) and open (connected) pores [19]. Moreover, high total porosity and the presence of non-controlled structural defects often result in poor mechanical properties of the final product. Thus, for almost all types of living tissues (connective, cartilaginous, bone, etc.), optimal scaffold porosity is usually considered to be in the range of 40% to 80%, with an average pore size of 50 to 700 microns [5,17].

A deep understanding of the internal architectonics of the biomedical product and controlled drug release carrier, based upon detailed structural data at scales ranging from the intermolecular interactions of the starting materials (ca. 1 nm) to the typical pore size of custom-designed implants or scaffolds for TEC (0.1–1 mm), is imperative to predict the behavior and changes in the physical, chemical and mechanical properties of such biomedical products over the course of their interaction with the surrounding biological medium.

Many analytical techniques and types of measuring equipment exist that enable the high-quality acquisition of the structural data mentioned above [20]. In the context of the present study, such techniques include: small-angle X-ray scattering (SAXS) [21], helium pycnometry (HP) [22,23], mercury intrusion porosimetry (MIP) [24,25] and X-ray microtomography (micro-CT) [26]. All these techniques, with some reservations (especially when we are talking about polymeric materials and structures), can be considered non-destructive. Each one has its own range of measurements, applications, and limitations.

In general, all fundamental properties of a material, one way or another, are related to the structure and arrangement of domains at the nanoscale. There are several classical methods, such as atomic force-, transmission electron- and scanning electron microscopy (TEM and SEM, respectively), which are used to characterize materials at this scale. However, SEM and TEM mainly allow for 2D measurements in a limited field of view and are, therefore, inadequate at investigating the pore interconnectivity of 3D specimens. These methods also have the disadvantage that the averaged results of a sample can rarely be obtained.

Ideally, SAXS is used in complement to microscopic methods since it provides representative structural information about a large sample area and its signal can be observed whenever a material contains structural features typically in the range of 1–100 nm. HP measures the true (absolute) volume and skeletal density of the sample. MIP has been routinely used to evaluate the pore-size distribution of powdered and bulk materials with open and interconnected pore structures at a scale ranging from a few nanometers to hundreds of micrometers. However, to analyze the “left end” of this range (pore diameter less than 10 nm), one needs to apply an intrusion pressure above 100 MPa. Furthermore, the destructive features of MIP cannot be ignored in this case, particularly for fragile or highly porous polymer scaffolds. In case of SAXS and micro-CT analysis it is always necessary to keep in mind X-ray intensity- and doze-dependent polymer degradation due to radiation-induced macromolecule scission.

Supercritical carbon dioxide (sc-CO_2_) swelling and plasticization of amorphous polymers, followed by the foaming of said polymers, represents just one of many methodologies for the fabrication of porous structures with a predetermined porosity [27]. Nevertheless, this approach has unambiguous advantages over conventional techniques for porous polymer production, such as solvent casting, particulate leaching, phase separation, and freeze-drying [28]. These advantages are based on its “solvent-free” nature, providing an absence of toxic organic solvent residues and retention of the bioactivity of the active and/or thermally labile components in the pharmaceutical formulations and bioactive scaffolds. Such qualities result from the fact that the critical temperature for carbon dioxide is *P_cr_* = 31 °C and all processes can be performed at near ambient (around 40 °C) temperatures [29,30,31].

Over the last two decades, almost all possible analytical techniques (except SAXS) have been applied to characterize both the surface and internal structures of porous biodegradable PLA and PLGA porous structures for TEC [32,33], including scaffolds fabricated with the use of supercritical carbon dioxide [34,35]. However, at least two questions still remain: (1) what kind of raw polymer internal structure transformation can one expect at the nanoscale after processing this material with sc-CO_2_; (2) how do the results collected with the help of these techniques complement and/or (arguably more importantly) correspond one to another?

The goals of the present work are to perform a SAXS analysis of the nanostructure transformation of polylactoglycolide, before and after processing with sc-CO_2_ and to comparatively study the internal structure assessment and porosity measurements for PLGA scaffolds (performed by SAXS, HP, MIP and micro-CT techniques).

## 2. Materials and Methods

Medical-grade polylactic-co-glycolic acid Purasorb PDLG7502 (Purac Biochem bv, Gorinchem, The Netherlands) with an inherent viscosity of C = 0.2 g/dL and a lacticto/glycolic acid monomer ratio of 75:25 was used as the raw polymer material. Chemically pure carbon dioxide (99.998% grade, NIIKM Ltd., Moscow, Russia) was applied as a plasticizing and foaming agent.

In our experiments, carbon dioxide gradually diffused into the polymer volume during the CO_2_ exposure time. The polymer swelled and intermolecular bonds were weakened. As a result, the polymer transformed into a liquid-like (plasticized) state and was ready for further foaming at the depressurization stage. Consequently, carbon dioxide plays a dual role as a plasticizing agent at the preliminary stage of the process and a foaming agent during the main stage (described in detail elsewhere [36]).

The initial PLGA granules (2 ÷ 3 mm in diameter) were ground by a rotational cryomill (LZM-1M, OLIS Ltd., Russia), followed by their consecutive sieving through a set of calibrated sieves (Grokhot GR30, Vibrotechnic Ltd., Saint Petersburg, Russia) with mesh sizes of 50 × 50 μm^2^ and 100 × 100 μm^2^ in order to select the fraction with particle sizes of ≤100 μm.

Next, a certain amount (ca. 600 mg) of finely dispersed PLGA powders was carefully loaded into a revolving type aluminum mold, comprising 12 cylindrical Teflon inserts (5 mm in diameter and 5 mm in height each). This mold was put into a high-pressure chamber which was further sealed and then purged (to remove atmospheric air), and gradually filled with carbon dioxide using a high-pressure pump (PN 101, NWA GmbH, Lörrach, Germany) at a predetermined pressure and temperature—10.0 MPa and 40 °C, respectively. The whole system was kept at these conditions for 1 h. After termination of the process, CO_2_ pressure was released up to the atmospheric value through the needle valve over the course of 30 s. Then, the molds and their contents were kept under room conditions for 24 h, as required for the complete removal of CO_2_ from the polymeric scaffolds and their final hardening. Porous cylindrical samples were removed from the mold inserts and put into plastic containers where they were stored at −18 °C until further analysis.

The morphology of the surface and internal structure of the scaffolds were studied via optical microscopy (performed with Bresser Advance ICQ stereoscopic microscope (Bresser, Broken, Germany) equipped with a Levenhook C510 camera (Levenhook, Tampa, FL, USA) and scanning electron microscopy ((SEM) using Phenom ProX scanning electron microscope (Phenom, Eindhoven, The Netherlands)). The accelerating voltage used to obtain the desired images was either 5 or 10 kV, with the magnification scale increasing from 50 to 1000. Prior to the studies, all samples were fixed to the microscope stage with conductive carbon tape, with no current-conducting (metallic) material having been deposited on them.

The measurements of the scaffold gravitational mass were carried out on VIBRA AF-R220CE balance (Shinko, Tokyo, Japan). The skeletal density of the polymer structures was measured by HP at room (20 °C) temperature using a pycnometer Pycnomatic ATC (Thermo Fisher Scientific, Milan, Italy).

Determination of the volume and pore size in the range of 1 to 100 nm was carried out by the small-angle X-ray scattering method on an automatic small-angle X-ray diffractometer “AMUR-K” [37] with a single-coordinate position-sensitive detector OD3M, at a fixed radiation wavelength *λ* equal to 0.1542 nm (CuKα line of a fine focus tube with a copper anode, a pyrolytic graphite monochromator) and a Kratky collimation system. The X-ray beam cross-section was 0.2 × 8 mm, and the scattering angle region corresponded to the range of the scattering vector modulus 0.1 < *s* < 1.0 nm^−1^ ($\left|s\right|\;=\;\frac{4\pi \cdot \sin\theta }{\lambda }$, 2*θ* is the scattering angle). The measurements were carried out in an evacuated chamber with a sample-detector distance of 700 mm. The measurement time for each sample was 20 min. Residual scattering by the empty chamber was subtracted from the scattering data of the samples. The measurement procedure was carried out according to the certified method [38].

To calculate the size distributions of inhomogeneities (in our case, pores dominated, which will here be called “particles”), we used MIXTURE software [39], which is freely available in the ATSAS small-angle scattering data processing software package [40], and the POLYMIX version of the software, which uses a rather more efficient minimization algorithm. As an alternative method, a direct nonparametric search for the distribution histogram from the scattering data (VOLDIS program) was used. In the first case, the scattering intensity from a multicomponent system of particles is represented as a linear combination of the contributions of scattering from *K* components, with each component being a system of particles with a (partial) size distribution described by the analytical expression:(1)$$I(s)\;=\;\sum_{k=1}^{K}\left[\rho_{k}^{2}v_{k}\right]\cdot I_{k}(s)$$
where *ν_k_* is the relative volume fraction of the *k*-th component, *ρ_k_* is the average contrast of the electron density (if only the shape of the distribution is of interest, then the contrast is assigned as being equal to 1.0), *I_k_*(*s*) is the scattering intensity of the *k*-th component, which depends on the type of the distribution function $D_{V_{k}}(r)$ and the given form factor of the particles $I_{0_{k}}(s,r)$:(2)$$I_{k}(s)\;=\;\int_{R_{\min}}^{R_{\max}}D_{V_{k}}(r)\cdot I_{0_{k}}(s,r)\cdot F_{k}(s)\cdot dr$$
where *r* is the particle size and $F_{k}(s)$ is the structure factor that takes into account interparticle interference in cases where they are densely packed. The result is a linear combination of partial distributions:(3)$$D_{V}(r)\;=\;\sum_{k=1}^{K}v_{k}D_{V_{k}}(r)$$

In MIXTURE software, to minimize the quadratic discrepancy between the experimental and theoretical (1) scattering curves, the variable metric method with simple restrictions on the parameters is used in the programs POLYMIX and VOLDIS and a modified version of the Levenberg–Marquardt algorithm is implemented. The unknowns are the parameters of the partial distributions (mean and half-width) and their contributions in Equation (3). The Schultz–Zimm distribution was used as the partial analytical expression, as described in [41].

Measurements of the scaffold volume and pore sizes in the range of 100 nm to 100 μm were carried out by MIP using the Pascal 140 (for determining the bulk density) and Pascal 240 (for determining the apparent density) (Thermo Fisher Scientific, Milan, Italy) porosimeters. Briefly, MIP is based on the properties of mercury as a non-wetting liquid in relation to many solid materials, due to which, mercury penetrates into the open pores of a solid sample with increasing pressure. The pore volume distribution is calculated from the experimental data on the amount of mercury that has penetrated into the pores of the sample and the equilibrium pressure at which the penetration phenomenon occurs. The calculations are usually based on the following assumptions: (1) the surface tension of mercury and the contact angle of the solid material are constant during the analysis; (2) the penetration pressure must be in equilibrium; (3) the pores are cylindrical; (4) solid materials do not deform under high pressure.

In the range of 100 µm to 1 mm, measurements of the pore volume and sizes of PLGA scaffolds were performed using X-ray microtomography. The tomographic method makes it possible to compute a three-dimensional voxel model of the object under study. This can then be used not only to estimate the integral porosity of the matrix (as traditional sorption methods do), but also to estimate the spatial distribution of the pores, as well as their size distribution. X-ray computer microtomography (micro-CT) measurements were performed on a laboratory microtomograph, TOMAS (at the Federal Scientific Research Center “Crystallography and Photonics” Russian Academy of Sciences) [42]. The values for the accelerating voltage and current were 40 kV and 20 mA, respectively. The probing energy was 17.5 keV (a pyrolytic graphite crystal was used as a monochromator). For each measurement, 400 radiographic projections were collected in the angular range of 200 degrees with a step of 0.5 degrees. The measurements were carried out according to the parallel scanning scheme. The XIMEA xiRAY11 detector used had a pixel size of 9 × 9 microns, and the total scanning time was 120 min. The spatial resolution was 15 μm. The algebraic coupled gradient method (CGLS) was used for tomographic reconstruction.

A preliminary analysis of porous PLGA samples—using synchrotron radiation phase-contrast tomography (SR-XPCT)—was performed at the Diamond-Manchester Imaging Branchline I13-2 [43,44] at the Diamond Light Source synchrotron, UK. A filtered polychromatic “pink” beam (8–30 keV) of parallel geometry was generated by an undulator with a 5 mm gap. For each tomogram, 4001 projection images were acquired at equally spaced angles over 180° of continuous rotation. Images were collected by a pco.edge 5.5 Camera Link detector (PCO AG, Germany) mounted on a visible light microscope of variable magnification. A 4× objective, coupled to a 500 μm LuAG:Ce scintillator, mounted ahead of a 2× lens, provided a total magnification of 8× *g*, a field of view of 2.1 × 1.8 mm (2560 × 2160 pixels) and an effective pixel size of 0.8125 μm. A propagation distance of approximately 390 mm was used. Data were reconstructed using a filtered back projection algorithm in Savu [45], incorporating flat- and dark-field corrections, optical distortion correction [46] and ring artefact suppression [47].

## 3. Results and Discussions

A typical general view and SEM images of the porous cylindrical PLGA scaffolds with a mean (averaged over 12 samples) weight of 0.0476 g, produced by sc-CO_2_ swelling, plasticization, and foaming are presented in Figure 1 and Figure 2, respectively.

### 3.1. Small-Angle X-ray Scattering (SAXS)

Small-angle scattering was used to assess the porosity of the cylinders and the initial polymer granules. The volume fraction of inhomogeneities in the large sizes region for the initial sample was significantly lower than for the porous cylinder images (Figure 3). It should be noted that the amplitude of the structure factor in Equation (2) did not exceed 2% of the scattering intensity, which indicates significant distances between the pores and, consequently, their closed nature. The distribution shapes are shown in Figure 3.

### 3.2. Helium Pycnometry and Mercury Intrusion Porosimetry

The results of the HP and MIP measurements of the PLGA scaffolds—the average values for the skeletal, volumetric (external volume), and apparent (calculated volume of penetrated mercury) densities—are presented in Table 1.

The apparent density is different from the skeletal density of the material. When comparing the porosity results measured by two different methods—helium pycnometry and mercury porosimetry—the values were 32.64% and 54.90%, respectively. The difference of 22.26% indicates the presence of closed pores that are inaccessible to helium and open up when a high pressure of ca. 20 MPa is applied. However, simultaneously with the opening of the closed pores, a strong compression of the material is observed (this is evidenced by the negative value of the difference in porosity measured by the two methods). It is not possible to accurately determine the proportion of closed pores due to sample compression.

Based on the mercury intrusion data (the dependence of the mercury volume inside the sample on the pressure applied) presented in Figure 4a (blue line), the specific pore volume, specific pore surface area, and average pore diameter (defined as four times the pore volume, divided by the surface area) were determined. Here, we assumed that all pores were cylindrical, and median pore diameter is defined as the pore size calculated at 50% of the total pore volume (Table 2).

Histograms of the pore size distribution in the volume of the scaffold and the specific surface area distribution by pore size are shown in Figure 4a,b, respectively.

### 3.3. X-ray MicroCT

To calculate the structural geometric parameters of the studied object, it is necessary to binarize the image, i.e., to separate the material matrix from the voids. Qualitative implementation of such a procedure is hindered by the measurement of noise. For this purpose, the “random walking” binarization technique was used [48]. This method requires one to define some points in the voids and in the matrix. Here, the 20th and 80th percentiles of pixel intensity as levels of the initial approximation of the voids and matrix have been applied. A small variation in these parameters slightly changes the result of binarization, showing that the algorithm is stable. A visualization of the binarized volume is shown in Figure 5.

The analysis of the binarized volume connectivity showed that almost all pores visible via computed tomography (over 10 µm in diameter) were open and connected. The volume fraction of closed pores was just 0.38%.

In Figure 6, it can be seen that pores, in general, have a convex shape. Therefore, in the tomography analysis, we used an approximation of the spherical pores, unlike the cylindrical approximation in mercury intrusion porosimetry, which is used for accounting for the connectivity of the pores.

To calculate the pore size distribution, taking into account the spherical pore approximation, the following algorithm was implemented:(1)for each point belonging to the pores, the Euclidean distance Ed(x,y,z) to the nearest matrix edge was calculated;(2)to determine the radii of the pores (approximating that the pores are spherical), the positions of local maxima of function Ed(x,y,z) were found (distance to the closest border) and these values used as the radii of the pores. The coordinates of the maxima were used as the centers inscribed in the spheres of the pores. This step is similar to the approach implemented in [49], in which the method was used to analyze the thickness of the wall, instead of the pore size, as is the case here;(3)the “watershed search” segmentation method [50], with the starting points in the positions of the local maxima identified earlier, was used for three-dimensional segmentation;(4)using a pseudo-random color scale, the three-dimensional segmented areas were colored (Figure 6);(5)for each segmented element, the volume and effective radius of the sphere was calculated.

It should be noted that X-ray microtomography, unlike other methods, makes it possible not only to obtain numerical characteristics, but also to conduct a total visual assessment of the sample structure and to detect various important features. For example, in our study, we found that the apparent anisotropy of the scaffold volume—expressed as the directional orientation of the polymer structures (Figure 7)—reflects the results of the internal stress of the polymer plasticized by sc-CO_2_, at the final stages of its foaming and solidification within the mold, during carbon dioxide pressure release.

Figure 8 shows a summary of the PLGA scaffold pore size distribution, calculated from SAXS HP, MIP, micro-CT, and SR XPCT measurements.

Figure 8 shows that the size range from 10 to 30 μm still remains uncertain. The combination of the methods allows us to estimate the fraction of open pores for the different pore sizes. However, it is likely that the application of high-resolution synchrotron microtomography can help to fill this gap.

To prove this concept, we performed various preliminary experiments at the diamond light source synchrotron to detect the pores inside our PLGA scaffolds with diameters of less than 30 µm, using a high-resolution detector in combination with phase-contrast imaging. The application of a high-resolution detector with an effective pixel size of 0.8 µm at the beamline I13-2 allows us to increase the spatial resolution, and thus, to visualize pores as small as 3 μm with a reasonable level of contrast (Figure 9). The spherical form of the micropores identified in the micro-CT measurements confirms the suggestion that in the analysis of small-angle scattering measurements, we can assume that pores have a spherical form.

To calculate the cumulative volume of the submicron pores under laboratory conditions, monochromatic tomography can be used to determine the linear absorption coefficient µ [51]. This parameter µ and the chemical formula of PLGA help to calculate the real density $\rho_{exp}$ of the PLGA. The relation $1\;-\;\frac{\rho_{exp}}{\rho_{tabulated}}$ (where $\rho_{tabulated}$ is the tabulated value of PLGA density) gives the cumulative volume of the submicron pores, which cannot be determined via tomography images.

## 4. Conclusions

A multiscale analysis of the polylactic-co-glycolic acid (Purasorb PDLG 7502) internal structure, before and after its supercritical fluid processing, was carried out using X-ray small-angle scattering, helium pycnometry, mercury intrusion porosimetry, X-ray microtomography, and—for the first time—the synchrotron light source. The proposed approach allows one to study the porosity of polymer scaffolds in the range of 0.02 to 1000 μm. Here, we have shown that X-ray small-angle scattering can be effectively used to study pores with a radius of less than 0.05 μm, while mercury porosimetry is not suitable for such small pore sizes since it requires high pressures that can destroy the fragile polymer walls. This approach opens up the possibilities for the wide-scale evaluation, computer modeling, and prediction of the physical and mechanical properties of PLGA scaffolds, as well as their biodegradation behavior in the body. Hence, this study has contributed to optimizing the process parameters of PLGA scaffold fabrication for specific biomedical applications.

## Figures and Tables

**Figure 1 polymers-13-01021-f001:**
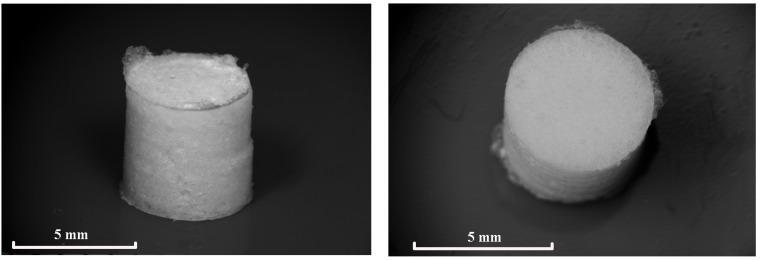
Optical images of PDLG 7502 porous samples.

**Figure 2 polymers-13-01021-f002:**
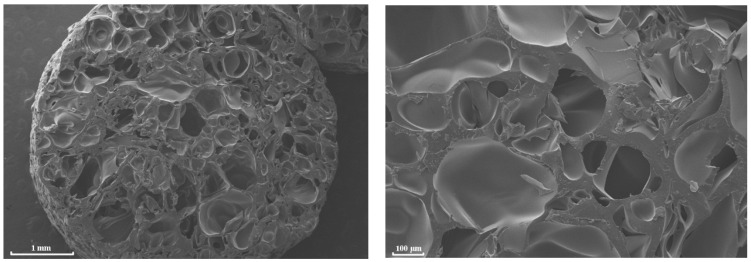
SEM images of PDLG 7502 porous samples at two different magnifications.

**Figure 3 polymers-13-01021-f003:**
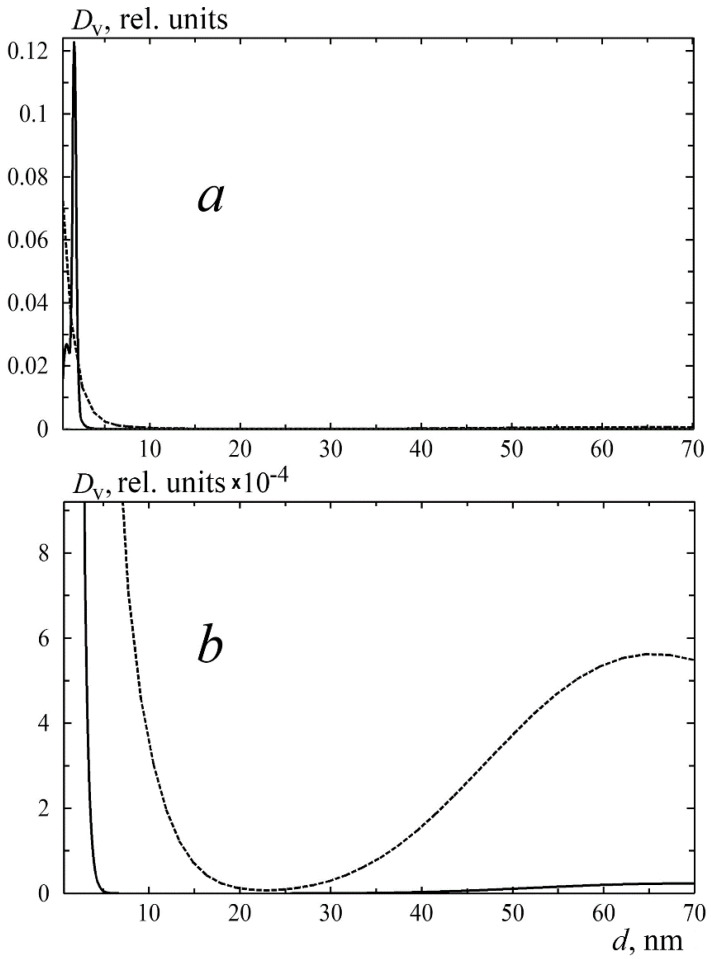
Volumetric distributions of inhomogeneities in the spherical approximation. The relative intensity of the distributions corresponds to the ratio of the scattering intensities. The data are shown in full scale (**a**), a large diameter region is enlarged (**b**). Dashed line—sc-CO_2_ processed sample; solid line—initial PLGA granule.

**Figure 4 polymers-13-01021-f004:**
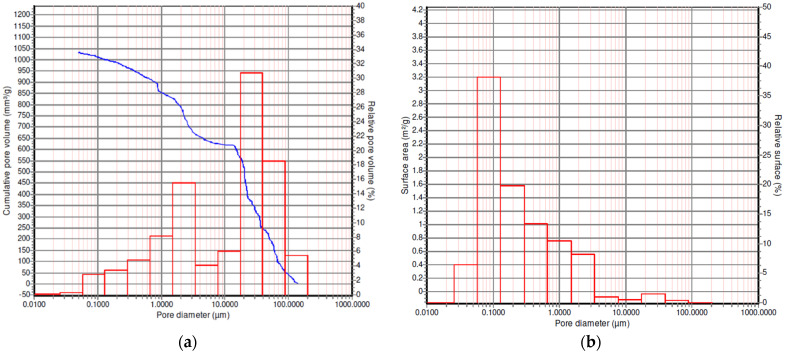
(**a**) Pore diameter distribution in the PLGA scaffold volume (blue line—curve of mercury intrusion into the sample volume at pressure ranging from 0 to 30 MPa); (**b**) distribution of specific surface area by pore sizes.

**Figure 5 polymers-13-01021-f005:**
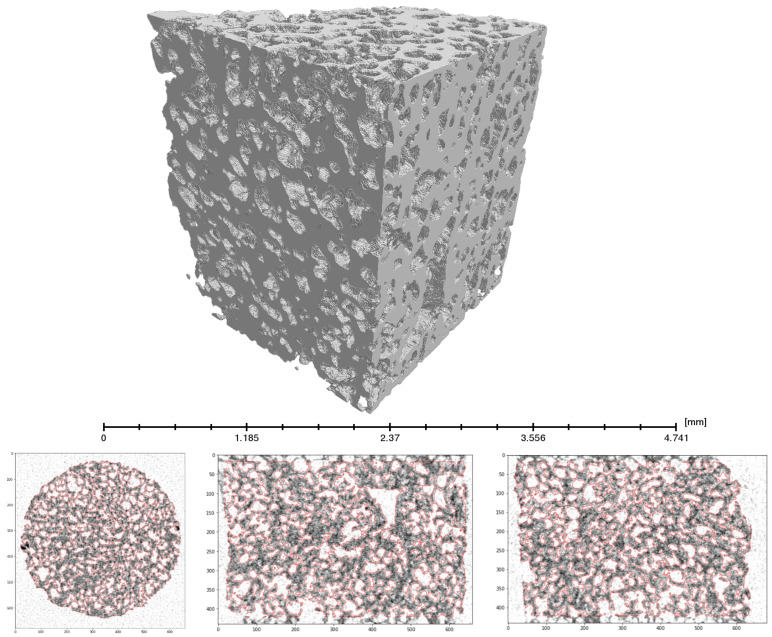
Example of binarization shown on three orthogonal reconstruction layers. Above: three-dimensional visualization of the binarized image. Bottom: gray image—tomographic reconstruction, red lines—calculated matrix boundaries.

**Figure 6 polymers-13-01021-f006:**
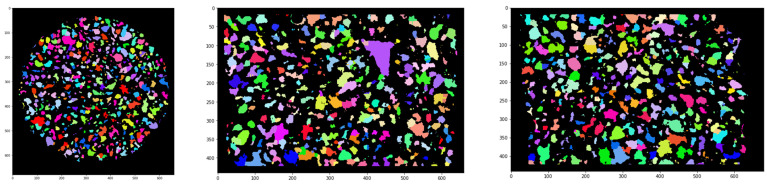
Orthogonal tomographic cross-sections of the pores in the spherical approximation. Each pore is colored in a unique color.

**Figure 7 polymers-13-01021-f007:**
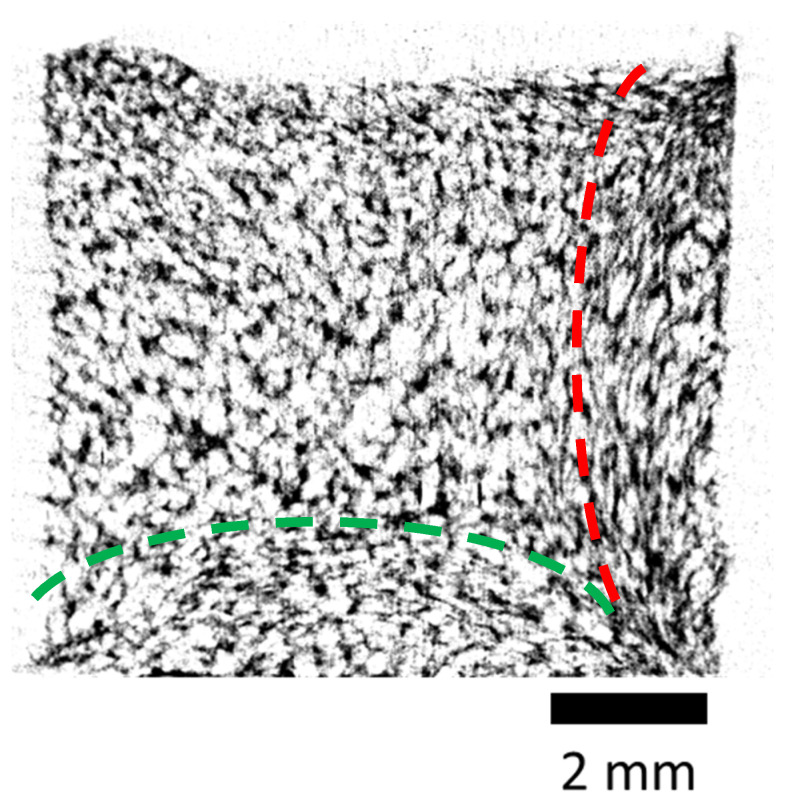
X-ray microtomography (micro-CT) images of PLGA scaffold cross-sections, demonstrating the presence of selected directions (anisotropy) in their structure (red and green dotted lines show the regions of high anisotropy).

**Figure 8 polymers-13-01021-f008:**
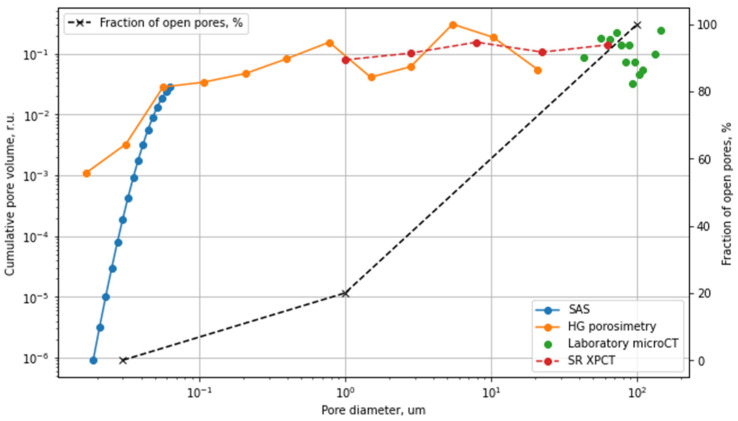
Cumulative pore size and the fraction of opened pores obtained by different methods.

**Figure 9 polymers-13-01021-f009:**
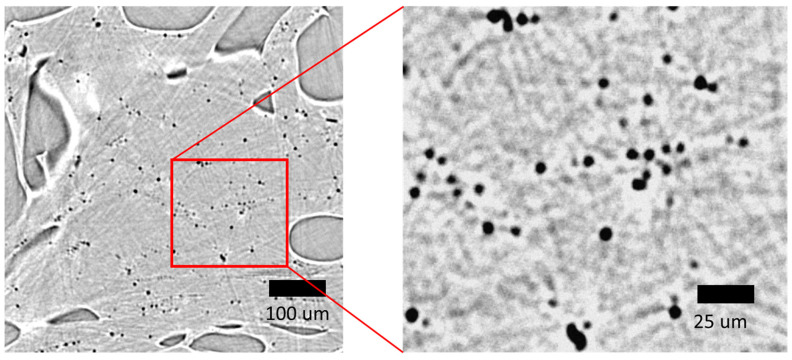
Synchrotron images of the cross-section of PLGA scaffold internal domains. The magnified image demonstrates the spherical form of the micro-pores. Image seems blurred due to the spatial resolution of micro-CT being approximately 3 μm, while the effective pixel size is 0.8 µm.

**Table 1 polymers-13-01021-t001:** Skeletal, bulk, and apparent densities, and mass of PDLG 7502 porous cylinders.

Sample Mass, g	Skeletal Density, g/cm^3^	Bulk Density, g/cm^3^	Apparent Density, g/cm^3^
0.0476	0.7891	0.5012	1.0353

**Table 2 polymers-13-01021-t002:** Porosity data of polylactic-co-glycolic acid (PLGA) scaffolds.

Specific Pore Volume, mm³/g	Specific Pore Surface Area, m²/g	Average Pore Diameter, μm	Median Pore Diameter, μm
1033.58	3.538	1.1686	20.2424

## Data Availability

The data presented in this study are available on request from the corresponding author.

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
