# Peer review of "Wide-Ranging Multitool Study of Structure and Porosity of PLGA Scaffolds for Tissue Engineering"

_polymers, 2021, doi:10.3390/polym13071021_

Round 1

Reviewer 1 Report

To the authors,

This manuscript is well organized and the methodology for studying the main goal of this work is properly designed. Using supercritical carbon dioxide for the generation of scaffolds for tissue engineering is been deeply studied. Here the authors could analyze a newly designed PLGA based scaffold made through this technique. To put in a nutshell, in my opinion, the authors could successfully prove their claims about the generation of a range of pores with different sizes in the scaffolds, however, there are some comments to improve the quality of the manuscript:

1- Polylactoglycolide is not the exact term for PLGA. PLGA stands for Poly Lactic-co-Glycolic Acid. This is a co-polymer and should be phrased in the correct way.

2- Each figure in Figures 1 and 2 should be separated.

3- Please check the English grammar and correct the small mistakes. 

4- Figure 9 should be named 9-a and 9-b.

5- Figure 9-a should is out of focus and should be corrected. The authors should provide images with higher quality.

Regards

Reviewer 2 Report

The manuscript “Wide-ranging multitool study of structure and porosity of PLGA scaffolds for tissue engineering” deals with the production of PLGA porous scaffolds by supercritical CO2 foaming. An intensive morphological characterization has been performed. However, the manuscript requires some improvements.

Detailed comments:

- Introduction. Among the technologies reported for the production of porous structures, also supercritical CO2 drying should be mentioned; indeed, this technology is frequently used to produce aerogels for tissue engineering applications. For example, see Baldino et al., Natural aerogels production by supercritical gel drying, Chemical Engineering Transactions, 2015, 43, pp. 739–744; etc...

- Materials and Methods. Please, separate materials used and scaffolds preparation procedure.

On which basis the selection of the foaming operative parameters (i.e., pressure, temperature and depressurization time) has been performed? Please, describe the possible effect of the process parameters on the scaffolds morphology.

- Results and Discussion. Since the produced PLGA scaffolds could be applied in tissue engineering, please, specify for what tissue the porosity and mean pore size are suitable.

- Please, check and correct some typing errors; also English has to be improved.

Round 2

Reviewer 2 Report

The authors performed the modifications proposed by the Reviewer and improved the manuscript.